

# Source contributions and potential reductions to health effects of particulate matter in India

Hao Guo[1], Sri Harsha Kota[2,3,4], Kaiyu Chen[1], Shovan Kumar Sahu[4], Jianlin Hu[2], Qi Ying[5], Yuan Wang[6], Hongliang Zhang[1,*]

[1]Department of Civil and Environmental Engineering, Louisiana State University, Baton Rouge LA 70803, USA

[2]Jiangsu Key Laboratory of Atmospheric Environment Monitoring and Pollution Control, Jiangsu Engineering Technology Research Center of Environmental Cleaning Materials, Nanjing University of Information Science & Technology, Nanjing 210044, China

[3]Department of Civil Engineering, Indian Institute of Technology Delhi, 110016, India

[4]Department of Civil Engineering, Indian Institute of Technology Guwahati, 781039, India

[5]Zachry Department of Civil Engineering, Texas A&M University, College Station, TX 77843, USA

[6]Division of Geological and Planetary Sciences, California Institute of Technology, Pasadena, California, 91106 USA

[*]Corresponding author: Hongliang Zhang. Email: hlzhang@lsu.edu. Phone: +1-225-578-0140





**Abstract**
Health effects of exposure to fine particulate matter ($PM_{2.5}$) in India were estimated in this study
based on a source-oriented version of the Community Multi-scale Air Quality (CMAQ) model.
Contributions of different sources to premature mortality and years of life lost (YLL) were
quantified in 2015. Premature mortality due to cerebrovascular disease (CEV) was the highest in
India (0.44 million), followed by ischaemic heart disease (IHD, 0.40 million), chronic obstructive
pulmonary disease (COPD, 0.18 million) and lung cancer (LC, 0.01 million), with a total of 1.04
million deaths. The states with highest premature mortality were Uttar Pradesh (0.23 million),
Bihar (0.12 million) and West Bengal (0.10 million). The highest total YLL was two years in Delhi,
and the Indo-Gangetic plains and east India had higher YLL (~ 1 years) than other regions. The
residential sector was the largest contributor to $PM_{2.5}$ concentrations (~ 40 μg/m$^3$), total premature
mortality (0.58 million), and YLL (~ 0.2 years). Other important sources included industry (~ 20
μg/m$^3$), agriculture (~ 10 μg/m$^3$), and energy (~ 5 μg/m$^3$) with their national averaged contributions
of 0.21, 0.12, and 0.07 million to premature mortality, and 0.12, 0.1, and 0.05 years to YLL.
Reducing $PM_{2.5}$ concentrations would lead to a significant reduction of premature mortality and
YLL. For example, premature mortality in Uttar Pradesh (including Delhi) due to $PM_{2.5}$ exposures
would be reduced by 79% and YLL would be reduced by 83% when reducing $PM_{2.5}$ concentrations
to 10 μg/m$^3$.
**Keywords:** Premature mortality, YLL, India, $PM_{2.5}$ exposure, CMAQ



## 1. Introduction

Due to insufficient control of emissions from a rapid increase in population, industries, urbanization and energy consumption, health effects associated with air pollution in developing countries in Asia are severe (Cohen et al., 2005). India, the second most populous country in the world, has been experiencing extremely high concentrations of fine particulate matter ($PM_{2.5}$) in recent decades. In 2015, $PM_{2.5}$ concentrations in south, east, north and west Indian cities were 6.4, 14.8, 13.2 and 9.2 times of the World Health Organization (WHO) annual guideline value of 10 $\mu g/m^3$ (Garaga et al., 2018). It is estimated that India accounted for 0.65 million out of the 3.3 million deaths resulted from air pollution caused by $PM_{2.5}$ globally in 2010 (Lelieveld et al., 2015). Outdoor $PM_{2.5}$ was also ranked as seventh in causes of death in India during 1990-2010 (IHME, 2013).

Efforts have been made to estimate the premature deaths associated with $PM_{2.5}$ in India. For example, Sahu and Kota (2017) estimated that 41 out of 100 thousand lives in Delhi could be saved by meeting the World Health Organization (WHO) suggested annual $PM_{2.5}$ guideline based on time series analysis. Such studies require extensive data, which is not available in all Indian cities. Few studies estimate the health effects using regional and global models, and satellite data. Lelieveld et al. (2015) estimated the global premature mortality of chronic obstructive pulmonary disease (COPD), cerebrovascular disease (CEV), ischaemic heart disease (IHD) and lung cancer (LC) using predicted $PM_{2.5}$ concentrations from a global atmospheric model and exposure-response equations from Burnett et al. (2014). In addition to premature mortality, years of life lost (YLL) an important indicator for health effects associated with $PM_{2.5}$, which accounts for the ages of those who die and age distribution of population, is more informative and meaningful for estimation of the burden of air pollution on health and environmental policy decision. Fann et al. (2012) used exposure risk functions from a cohort study by American Cancer Association (Krewski et al., 2009) and life expectancy and life lost with standards tables from Centers of Disease Control to estimate nearly 1.1 million life years lost due to $PM_{2.5}$ exposure in 2005 in the United States. Ghude et al. (2016) predicted 0.57 million premature deaths and 3.4 ±1.1 years of YLL associated with $PM_{2.5}$ in India for 2011.

To effectively design pollution control strategies, the contributions of different emission sources to $PM_{2.5}$ concentrations are crucial. Source-oriented chemical transport models (CTM) based on



tagged tracer technique have been developed and used for source apportionment of gases (Kota et
al., 2014) and PM (Kota et al., 2015; Ying et al., 2015; Zhang and Ying, 2010) in the past. Guo et
al. (2017), which was the first study to use the source-oriented Community Multi-scale Air Quality
(CMAQ) model in India, showed residential sector contributed the most (~ 80 μg/m$^3$) to total PM$_{2.5}$,
followed by industry sector (~ 70 μg/m$^3$) in 2015. Recently, Hu et al. (2017) estimated the
premature mortality caused by different sources of PM$_{2.5}$ in China and showed that industrial and
residential sources contributed to 0.40 (30.5%) and 0.28 (21.7%) million premature deaths,
respectively. However, no studies have attributed the health effects to different sources of PM$_{2.5}$
in India till date.
The objective of this study is to estimate contributions of each emission sectors to PM$_{2.5}$ related
mortality and YLL in India. The potential health benefits of reducing PM$_{2.5}$ concentrations in
different Indian states are also explored. Such study would be of tremendous value for the
government to channel their resources in reducing pollution in India.
**2. Method**
**2.1 Model application for PM$_{2.5}$ prediction and source apportionment**
The models used in this study were based on CMAQ 5.0.1 with a modified SAPRC11
photochemical mechanism and aerosol module version 6 (AERO6). Heterogeneous formation of
SO$_4$, NO$_3$, and SOA formation from surface uptakes was incorporated to improve model
performance (Hu et al., 2016; Ying et al., 2015). Source contributions of primary PM (PPM) and
its chemical components were estimated using tagged non-reactive tracers. The tracers from each
source sector go through all atmospheric processes similar to other species. Detailed information
on this source apportionment method could be found in Guo et al. (2017) and the references therein.
The source contributions to secondary inorganic aerosol (SIA) were determined by tracking SO$_2$,
NOx, and NH$_3$ through atmospheric processing using tagged reactive tracers. Both the
photochemical mechanism and aerosol module were expanded so that SO$_4$, NO$_3$, and NH$_4$ and
their precursors from different sources are tracked separately throughout the model calculations
(Qiao et al., 2015; Zhang et al., 2014; Zhang et al., 2012).
The default vertical distributions of concentrations that represented clean continental conditions
provided by the CMAQ model were used for the 36-km domain covering the whole India (Figure
S1). The Weather Research & Forecasting model (WRF) v3.7.1 was utilized to generate





meteorology inputs for CMAQ, and Emissions Database for Global Atmospheric Research
(EDGAR) version 4.3 (http://edgar.jrc.ec.europa.eu/overview.php?v=431) were used for six
anthropogenic emissions: energy, industry, residential, on-road, off-road and agriculture. The
biogenic emissions were generated by Model for Emissions of Gases and Aerosols from Nature
(MEGAN) v2.1 (Guenther et al., 2012) and wildfire emissions were from the Fire Inventory from
NCAR (FINN), which was based on satellite observations (Wiedinmyer et al., 2011). Dust and sea
salt emissions were generated in line during simulations. Details of the model application and the
performance in 2015 can be found in Kota et al. (2018).
**2.2 Estimation of premature mortality**
The relative risk (RR) due to COPD, CEV, IHD and LC related mortality associated with long-
term exposure of $PM_{2.5}$ concentrations is calculated using integrated exposure-response function
estimated by Burnett et al. (2014) as described in Eq. (1) and Eq. (2).
$$RR = 1, \quad for\ c < c_{cf} \tag{1}$$
$$RR = 1 + \alpha\left\{1 - \exp\left[-\gamma\left(c - c_{cf}\right)^{\delta}\right]\right\}, \quad for\ c \geq c_{cf} \tag{2}$$
where $C_{cf}$ is the threshold concentration below which there is no additional risk. A total of 1000
sets of α, γ, δ and $C_{cf}$ values generated using Monte Carlo simulations for each disease were
obtained       from       the       Global       Health       Data       Exchange       website
(http://ghdx.healthdata.org/sites/default/files/record-attached-
files/IHME_CRCurve_parameters.csv). C is the predicted $PM_{2.5}$ concentration. RR values are
calculated for each set of α, γ, δ and $C_{cf}$ for all people above the age of 25 and for each grid cell in
the domain. Then, the premature mortality is calculated as Eq. (3).
$$\Delta Mort = y_o[(RR - 1)/RR]Pop \dots\dots\dots\dots\dots\dots\dots\dots\dots\dots\dots\dots\dots\dots\dots\dots\dots\dots\dots\dots \tag{3}$$
where $y_o$ refers to baseline mortality rate for a particular disease in India as listed in Table S1,
obtained from based on the WHO Mortality Database and Pop is the population in a certain grid
cell as listed in Table S2. The mean, lower (2.5%) and upper (97.5%) limits of premature mortality
associated with each disease in a grid are estimated using the 1000 RR values. Total premature
mortality is calculated by adding premature mortality for each disease in a grid. Total average
premature mortality in a state is obtained by adding all average premature mortalities of all grids





in the state multiplied by the fraction of the grid inside the state. A similar approach is used for
calculating the upper and lower limits of premature mortality.

**2.3 Estimation of years of life lost**

Years of life lost (YLL) is another important index to reflect the health impact of $PM_{2.5}$
concentrations (Guo et al., 2013; Pope III et al., 2009; Romeder and McWhinnie, 1977; Yim and
Barrett, 2012). It is a measure of the average years a person would have lived if he or she had not
died prematurely due to some specific reason. YLL is usually calculated as a summation of the
number of deaths at each age group multiplied by the number of years remaining as shown in Eq.

135    (4).

$$YLL = \sum_{i=1}^{n-1} a_i \, d_i = \sum_{i=1}^{n-1}(n - y(i) - 0.5) \, d_i \dots\dots\dots\dots\dots\dots\dots\dots\dots\dots\dots\dots\dots (4)$$
where $d_i$ is the number of deaths in age group i (i = 1,7) as shown in Table S2 n is the life
expectancy of India (male= 66.2 and female= 69.1 in 2013), y(i) is the mean age of age group i
and $a_i$ is the remaining years of life left when death occurs in age group i. In this study, the overall
YLL was divided by population in a certain grid cell to get life expectancy loss per person (Pope
III et al., 2009).

**3. Results**

**3.1 Predicted premature mortality and YLL**

Figure 1 shows the predicted annual $PM_{2.5}$ concentrations in India for 2015, with the highest
concentration of ~120 μg/m$^3$ in Delhi and some states in east India. The spatial distribution of
$PM_{2.5}$ concentration shows that the Indo-Gangetic plains have a higher concentration than other
regions. East and parts of central India also have high $PM_{2.5}$ concentrations, while west and south
India are less polluted. The population-weighted concentration (PWC) throughout the country is
32.8 μg/m$^3$ (Table 1). East India is the most polluted with 47.8 μg/m$^3$, closely followed by north
India 43.1 μg/m$^3$. PWC values are 31.2 μg/m$^3$ in south, 25.4 μg/m$^3$ in the northeast, 23.9 μg/m$^3$ in
the west and 23.5 μg/m$^3$ in central India. Delhi is the state with the highest PWC of 66.3 μg/m$^3$.
The states apart from Delhi, where PWC is higher than the national average, are Sikkim 54.7 μg/m$^3$,
West Bengal 54.1 μg/m$^3$, Bihar 53.1 μg/m$^3$, Haryana 47.3 μg/m$^3$, Uttar Pradesh 47.3 μg/m$^3$,
Jharkhand 39.2 μg/m$^3$ and Punjab 35.5 μg/m$^3$.



The total premature mortality for adults (≥ 25 years old) and those due to COPD, LC, IHD, and
CEV are also shown in Figure 1. The total premature mortality peaks at populous megacities
located at coastal area, Indo-Gangetic plains, and west India. For example, in Indo-Gangetic plains,
where the population density is more than 1 million per gird (i.e., 36 km×36 km), premature
mortality can be as high as 3000 deaths per 100,000 persons. Premature mortalities of COPD, LC,
IHD, and CEV show a similar spatial distribution with the total. CEV is the largest contributor and
has peak values at Indo-Gangetic plains. COPD and IHD are also important with a peak of ~ 1400
deaths per 100,000 persons at Indo-Gangetic plains. LC contributes the least to total premature
mortality.
Table 1 also shows that the total premature mortality for adults in India for 2015 is approximately
1.04 million with CI95 of 0.53-1.54 million. High premature mortality is in the populous states
such as Uttar Pradesh (0.23 million), Bihar (0.12 million) and West Bengal (0.10 million) as shown
in Figure S2. In addition, states such as Maharashtra (0.09 million) and Andhra Pradesh (0.06
million) also have high premature mortality. Generally, the states in Indo-Gangetic plains and east
India have a higher premature mortality than other states. South states have lower premature
mortality. Premature mortality due to CEV is highest in India (0.44 million), followed by IHD
(0.43 million), COPD (0.18 million) and LC (0.01 million) (Table 1). States with high PWC have
slightly higher CEV premature mortality compared to IHD. IHD and CEV constitute about 81 %
of the total premature mortality over the country in 2015.
Table S3 shows the comparison of the results with other studies. This study predicted higher total
premature mortality (1.04 million) compared to Lelieveld et al. (2015) (0.65 million) and Ghude
et al. (2016) (0.57 million). Considering the uncertainty range (0.53- 1.54 million), our result is
comparable with these two studies. The difference may be caused by the higher resolution (36 km)
compared with Lelieveld et al. (2015) (100 km) and different simulation episode (2015) compared
with Ghude et al. (2016) in 2011. The ratios of the four diseases to the total are close in this study
and Lelieveld et al. (2015), except IHD and CEV.
Figure 2 shows the total YLL and to the contributions of COPD, LC, IHD, and CEV. The YLL for
entire India is the highest for CEV (0.8 years) and closely followed by IHD (0.7 years). LC has
the least YLL (0.03 years), while COPD has the YLL of 0.45 years. YLL for states in north, east,
south and west India are 1.2, 1.0, 0.2 and 0.4 years, respectively. The highest total YLL is ~ 2



years in Delhi, indicating $PM_{2.5}$ concentrations strongly threaten the health of people living in the
capital of India. Indo-Gangetic plains and east India have higher YLL (~ 1 years) compared to
other regions. Another study conducted in India for 2011 showed that $PM_{2.5}$ concentration
associated lost life expectancy is $3.4 \pm 1.1$ years (Ghude et al., 2016). The difference is due to the
different episodes and methods in calculating YLL. In Ghude et al (2016), YLL was calculated
based on the linear relationship assumption that an increase of 1 $\mu g/m^3$ in $PM_{2.5}$ exposure decreases
mean life expectancy by about $0.061 \pm 0.02$ years (Pope III et al., 2009), which introduced
additional uncertainties to their result.
**3.2 Source apportionment of premature mortality and YLL**
Figure 3 shows the annual contributions of different sources to total $PM_{2.5}$ concentration.
Residential sector contributes highest to total $PM_{2.5}$ with ~ 40 $\mu g/m^3$, followed by industry sector
(~20 $\mu g/m^3$). Energy sectors and agriculture sector contribute to ~5 $\mu g/m^3$ and ~8 $\mu g/m^3$. In north,
east, south and west India, residential sector (~ 40 $\mu g/m^3$), residential sector (~ 15 $\mu g/m^3$),
residential sector (~ 5 $\mu g/m^3$) and industry sector (~ 30 $\mu g/m^3$) have the maximum contributions
to total $PM_{2.5}$, respectively. Open burning has significant high contributions (~ 1 $\mu g/m^3$) in
northeast India. Energy $PM_{2.5}$ concentrations have significant high concentration point at north (~
30 $\mu g/m^3$) and east (~ 15 $\mu g/m^3$) India compared to other parts of the country as several coal-based
power plants located there (Guttikunda and Jawahar, 2014). On the contrary, industry, residential
and agriculture sector distribute evenly at Indo-Gangetic plain. Residential source peaks in north
Pakistan and dust source peaks in desert areas in other countries. In most states, residential is the
largest contributor because residential heating during October to December are the main sources
of $PM_{2.5}$ (Vadrevu et al., 2011).
The total premature mortality due the eight source sectors and SOA is shown in Figure 4 and
portions of the contribution of each source type of each state in India is listed in Table S4.
Residential (55.45%), Industry (19.66%), Agriculture (11.90%), and Energy (6.80%) are the major
sources contributing to premature mortality due to $PM_{2.5}$ concentrations. Contributions of
residential, industry, agriculture and energy sectors are maximum in Bihar (62.01%), Delhi (40%),
Assam (24.37%) and Chhattisgarh (22.63%), respectively. Overall premature mortality in more
than 90% of the states is dominated by residential source. The uses of primitive methods of cooking
instead of cooking gas and electric heaters could be a top factor. Burning of solid fuels for cooking





and other purposes could be another important factor. Highest contributions to premature mortality
from residential sources are in states at Indo-Gangetic plains and east India. Premature mortality
of residential sector in south Indian states is lower compared with other parts of India, while
premature mortality of industry sector is more important in western states. Delhi is affected the
most among all states by industrial source, and premature mortality due to the energy sector is
higher in mineral-rich states such as Chhattisgarh. Agriculture $PM_{2.5}$ contributes highest to
premature mortality in Assam. Premature mortality in other northeast states such as Meghalaya,
Mizoram, Tripura, Manipur, Nagaland, and Sikkim are also contributed significantly by
agriculture $PM_{2.5}$. In comparison with Lelieveld et al. (2015), this study predicts higher
contributions from industry and agriculture sectors but lower from traffic and dust sectors due to
the differences in emissions (Table S3).
Figure 5 showed YLL attributed to different source types and SOA. Similar to the pattern of
premature mortality in Figure 4, residential is the top factor, which reduces ~ 0.6 years in severe
polluted and populous area like Delhi, followed by industry, energy, and SOA. A significant peak
of industry YLL is at west India and high YLL occurs at Indo-Gangetic plains. Unlike the spatial
distribution of industry contributions to YLL, YLL for energy sector shows some point sources of
energy emission in central India. For SOA, YLL is ~ 0.1 years for majority parts of India with a
high YLL (~ 0.35 year) in southeast India. YLL for agriculture sector distributes evenly at Indo-
Gangetic plains and peaks at west India (~ 0.12 year).
**3.3 Potential reduction of premature mortality with reduced $PM_{2.5}$ concentrations**
Figure 6 shows the normalized premature mortality with a fractional reduction in $PM_{2.5}$
concentrations (relative to 2015 concentrations) for the whole of India and top $PM_{2.5}$ polluted states,
Bihar, Maharashtra, Uttar Pradesh (including Delhi), West Bengal. It shows that the decrease of
premature mortality is slower in the beginning when $PM_{2.5}$ concentrations are higher, and the
marginal benefit of $PM_{2.5}$ reduction to premature mortality increases as PM concentrations
decrease. A 30% of reduction in $PM_{2.5}$ in whole India only lead to a 25% reduction in mortality
from the 2015 level without considering population increases, but 90% reduction in mortality
could be achieved with an 80% decreasing in $PM_{2.5}$. $PM_{2.5}$ concentrations need to be reduced by
65%, 50%, 60% and 65%, respectively, for Bihar, Maharashtra, Uttar Pradesh (including Delhi)
and West Bengal to achieve a 50% reduction in $PM_{2.5}$-related premature mortality.



Figure 7 evaluates the premature mortality and YLL benefit when PM$_{2.5}$ concentrations in the
whole of India and top PM$_{2.5}$ polluted states, Bihar, Maharashtra, Uttar Pradesh (including Delhi)
and West Bengal are reduced to four different standards, i.e., Indian National Ambient Air Quality
Standard (INAAQS) of 40 μg/m$^3$, WHO interim target 3 (WHO IT3) of 15 μg/m$^3$, the United
States (U.S.) Ambient Air Quality Standards (NAAQS) annual standard of 12 μg/m$^3$, and the WHO
guideline level of 10 μg/m$^3$. The reductions of the premature mortality when PM$_{2.5}$ concentrations
in the highly polluted regions (annual average concentration $\geq 40\,\mu g/m^3$) are shown in Table S5.
For example, the premature mortality in Uttar Pradesh (including Delhi) due to PM$_{2.5}$ exposure
will be reduced by 79% from 0.25 million to approximately 0.06 million and the YLL will be
reduced by 83% from 1.27 year to 0.22 year when PM$_{2.5}$ concentrations drop to 10 μg/m$^3$. The
reductions of premature mortality are also more significant in most populous states such as Uttar
Pradesh (79%) and West Bengal (80%). However, the decrease is not significant when PM$_{2.5}$
concentrations drop to current INAAQS standards for 40 μg/m$^3$ as it only reduces premature
mortality by 13.10% and YLL by 9.85% for the whole India. When PM$_{2.5}$ concentrations drop to
15 μg/m$^3$, premature morality for India will reduce to 0.37 million and YLL will decrease to 0.56
year. In 12 μg/m$^3$ case, premature mortality and YLL will be reduced to 0.17 million and 0.39 year.
This indicates that the current INAAQS standards are not sufficient to reduce health impacts of air
pollution in India.

### 263    4. Conclusion

A source-oriented CMAQ modeling system with meteorological inputs from the WRF model was
used to quantify source contributions to concentrations and health effects of PM$_{2.5}$ in India for
2015. The predicted annual PM$_{2.5}$ concentrations in India for 2015 could reach 120 μg/m$^3$ in Delhi
and some states in east India has a total mortality greater than 3000 deaths per 100,000 persons.
The total premature mortality in India for adult $\geq$ 25 years old in 2015 was approximately 1.04
million. Uttar Pradesh (0.23 million), Bihar (0.12 million) and West Bengal (0.10 million) had
higher premature mortality compared to other states. YLL peaks at Delhi with ~ 2 years and Indo-
Gangetic plains and east India have high YLL (~ 1 years) compared to other regions in India. The
residential sector is the top contributor (55.45%) to total premature mortality and contributes to ~
0.2 years to YLL with source contribution of ~ 40 μg/m$^3$ to total PM$_{2.5}$. Reducing the PM$_{2.5}$
concentrations to the WHO guideline value of 10 μg/m$^3$ would result in a 79% reduction of
premature mortality and 83% reduction of YLL in Utter Pradesh (including Delhi) due to PM$_{2.5}$





exposures. The total mortality and YLL of whole India would also be significantly reduced by
decreasing current $PM_{2.5}$ level to 10 μg/m$^3$.
**Acknowledgment**
Portions of this research were conducted with high performance computing resources provided by
Louisiana State University (http://www.hpc.lsu.edu). The project is funded by the Competitiveness
Subprogram (RCS) from Louisiana Board of Regents (LEQSF(2016-19)-RD-A-14). H.J. would
like to thank the support from the National Natural Science Foundation of China (41675125), and
Natural Science Foundation of Jiangsu Province (BK20150904), Jiangsu Six Major Talent Peak
Project (2015-JNHB-010).



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



Table 1. Population ($\times 10^6$), population-weighted concentration (PWC, μg/m$^3$) and premature mortality ($\times 10^4$ deaths) due to COPD, LC, IHD, and CEV in each state or union territory in India.

| State | Population | PWC | COPD | LC | IHD | CEV | Total |
|---|---|---|---|---|---|---|---|
| Andhra Pradesh | 85.3 | 22.45 | 0.96 (0.37, 1.63) | 0.07 (0.01, 0.11) | 2.48 (1.73, 3.54) | 2.18 (0.83, 3.42) | 5.69 (2.94, 8.70) |
| Arunachal Pradesh | 2.2 | 10.08 | 0.01 (0.00, 0.02) | 0.00 (0.00, 0.00) | 0.03 (0.02, 0.05) | 0.01 (0.01, 0.03) | 0.05 (0.03, 0.09) |
| Assam | 28.5 | 23.86 | 0.34(0.13, 0.57) | 0.02 (0.01, 0.04) | 0.86 (0.61, 1.23) | 0.80 (0.30, 1.25) | 2.03 (1.04, 3.09) |
| Bihar | 103.2 | 53.06 | 2.25 (1.08, 3.33) | 0.17 (0.05, 0.24) | 4.10 (3.14, 7.05) | 5.63 (1.79, 6.90) | 12.15 (6.07, 17.52) |
| Chandigarh | 0.2 | 30.51 | 0.00 (0.00, 0.01) | 0.00 (0.00, 0.00) | 0.01 (0.00, 0.01) | 0.01 (0.00, 0.01) | 0.02 (0.01, 0.03) |
| Chhattisgarh | 25.8 | 25.75 | 0.33 (0.13, 0.55) | 0.02 (0.01, 0.04) | 0.81 (0.58, 1.17) | 0.80 (0.29, 1.26) | 1.97 (1.01, 3.01) |
| Dadra & Nagar Haveli | 0.5 | 20.91 | 0.00 (0.00, 0.01) | 0.00 (0.00, 0.00) | 0.01 (0.01, 0.02) | 0.01 (0.00, 0.02) | 0.03 (0.02, 0.04) |
| Daman & Diu | 0.1 | 19.6 | 0.00 (0.00, 0.00) | 0.00 (0.00, 0.00) | 0.00 (0.00, 0.01) | 0.00 (0.00, 0.01) | 0.01 (0.00, 0.01) |
| Goa | 1.9 | 18.11 | 0.02 (0.01, 0.03) | 0.00 (0.00, 0.00) | 0.05 (0.04, 0.07) | 0.04 (0.02, 0.06) | 0.11 (0.06, 0.16) |
| Gujrat | 62.4 | 18.53 | 0.57 (0.21, 1.01) | 0.04 (0.01, 0.07) | 1.61 (1.07, 2.27) | 1.19 (0.48, 1.95) | 3.42 (1.77, 5.30) |
| Haryana | 37.4 | 47.32 | 0.75 (0.35, 1.13) | 0.06 (0.02, 0.08) | 1.43 (1.08, 2.39) | 1.88 (0.61, 2.38) | 4.12 (2.06, 5.98) |
| Himachal Pradesh | 8.8 | 15.08 | 0.06 (0.02, 0.11) | 0.00 (0.00, 0.01) | 0.18 (0.12, 0.26) | 0.12 (0.05, 0.20) | 0.37 (0.19, 0.58) |
| Jammu & Kashmir | 12.4 | 9.80 | 0.04 (0.01, 0.09) | 0.00 (0.00, 0.01) | 0.16 (0.08, 0.26) | 0.06 (0.02, 0.14) | 0.27 (0.11, 0.50) |
| Jharkhand | 36.4 | 39.25 | 0.65 (0.29, 1.00) | 0.05 (0.01, 0.07) | 1.33 (0.99, 2.14) | 1.66 (0.54, 2.20) | 3.68 (1.82, 5.41) |
| Karnataka | 63.0 | 16.23 | 0.51 (0.18, 0.94) | 0.04 (0.01, 0.06) | 1.56 (1.04, 2.12) | 0.97 (0.45, 1.55) | 3.08 (1.67, 4.67) |
| Kerala | 35.3 | 19.44 | 0.34 (0.12, 0.59) | 0.02 (0.00, 0.04) | 0.93 (0.63, 1.33) | 0.73 (0.29, 1.18) | 2.03 (1.05, 3.14) |
| Madhya Pradesh | 77.9 | 22.62 | 0.89 (0.34, 1.51) | 0.06 (0.01, 0.11) | 2.32 (1.65, 3.22) | 2.06 (0.82, 3.26) | 5.35 (2.81, 8.10) |
| Maharashtra | 117.1 | 28.61 | 1.58 (0.65, 2.57) | 0.11 (0.03, 0.18) | 3.72 (2.68, 5.44) | 3.73 (1.38, 5.52) | 9.14 (4.74, 13.70) |
| Manipur | 2.7 | 21.13 | 0.03 (0.01, 0.05) | 0.00 (0.00, 0.00) | 0.08 (0.05, 0.11) | 0.06 (0.03, 0.10) | 0.17 (0.09, 0.26) |
| Meghalaya | 4.3 | 22.07 | 0.05 (0.02, 0.08) | 0.00 (0.00, 0.01) | 0.13 (0.09, 0.17) | 0.11 (0.04, 0.17) | 0.29 (0.15, 0.43) |
| Mizoram | 1.5 | 19.72 | 0.02 (0.01, 0.03) | 0.00 (0.00, 0.00) | 0.04 (0.03, 0.06) | 0.03 (0.01, 0.05) | 0.09 (0.05, 0.14) |
| Nagaland | 3.2 | 19.51 | 0.03 (0.01, 0.06) | 0.00 (0.00, 0.00) | 0.09 (0.06, 0.12) | 0.07 (0.03, 0.11) | 0.19 (0.10, 0.29) |
| Delhi | 8.1 | 66.28 | 0.21 (0.10, 0.29) | 0.02 (0.01, 0.02) | 0.34 (0.27, 0.61) | 0.49 (0.16, 0.57) | 1.06 (0.54, 1.50) |
| Odisha | 43.4 | 29.59 | 0.63 (0.26, 1.01) | 0.05 (0.01, 0.07) | 1.44 (1.05, 2.17) | 1.57 (0.54, 2.32) | 3.69 (1.86, 5.57) |
| Puducherry | 1.2 | 15.40 | 0.01 (0.00, 0.02) | 0.00 (0.00, 0.00) | 0.03 (0.02, 0.04) | 0.02 (0.01, 0.03) | 0.05 (0.03, 0.08) |
| Punjab | 28.9 | 35.46 | 0.48 (0.21, 0.75) | 0.04 (0.01, 0.05) | 1.02 (0.75, 1.61) | 1.22 (0.40, 1.66) | 2.75 (1.37, 4.07) |
| Rajasthan | 71.4 | 20.86 | 0.74 (0.28, 1.28) | 0.05 (0.01, 0.09) | 2.00 (1.39, 2.80) | 1.64 (0.67, 2.54) | 4.44 (2.35, 6.71) |
| Sikkim | 4.5 | 54.72 | 0.09 (0.05, 0.13) | 0.01 (0.00, 0.01) | 0.16 (0.12, 0.29) | 0.22 (0.07, 0.26) | 0.48 (0.24, 0.69) |
| Tamil Nadu | 70.2 | 13.82 | 0.45 (0.15, 0.87) | 0.03 (0.00, 0.06) | 1.47 (0.88, 2.13) | 0.77 (0.33, 1.38) | 2.72 (1.36, 4.44) |
| Tripura | 3.7 | 26.04 | 0.05 (0.02, 0.08) | 0.00 (0.00, 0.01) | 0.12 (0.08, 0.17) | 0.12 (0.04, 0.19) | 0.29 (0.15, 0.44) |
| Uttar Pradesh | 211.2 | 47.19 | 4.26 (1.98, 6.41) | 0.32 (0.09, 0.45) | 8.10 (6.14, 13.63) | 10.80 (3.45, 13.59) | 23.48 (11.66, 34.09) |
| Uttarakhand | 11.9 | 15.04 | 0.08 (0.03, 0.14) | 0.01 (0.00, 0.01) | 0.23 (0.14, 0.33) | 0.16 (0.06, 0.26) | 0.47 (0.24, 0.74) |
| West Bengal | 88.9 | 54.13 | 1.93 (0.94, 2.86) | 0.14 (0.04, 0.20) | 3.51 (2.68, 6.00) | 4.75 (1.53, 5.81) | 10.34 (5.20, 14.87) |
| India | 1254.0 | 32.78 | 18.36 (7.94, 29.14) | 1.34 (0.35, 2.05) | 40.36 (29.22, 62.78) | 43.94 (15.27, 60.36) | 103.99 (52.78, 154.34) |



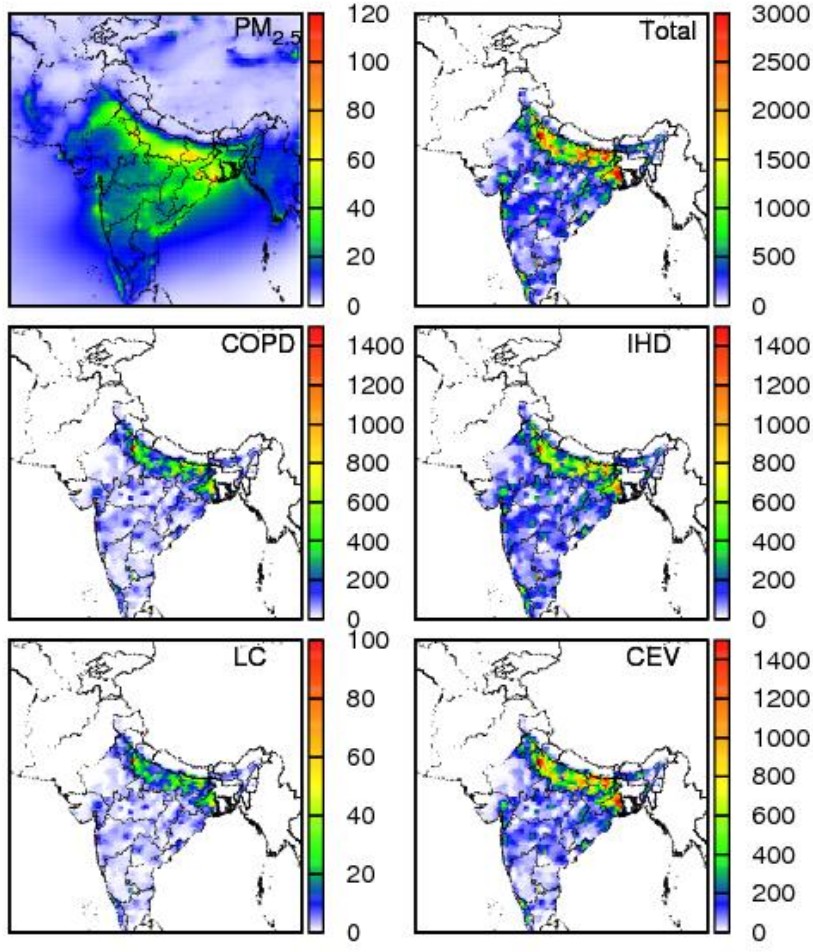

Figure 1. Predicted annual PM$_{2.5}$ concentrations (µg/m$^3$), total premature mortality (death per grid of $36 \times 36$ km$^2$) and premature mortality due to COPD, LC, IHD and CEV in India for 2015.

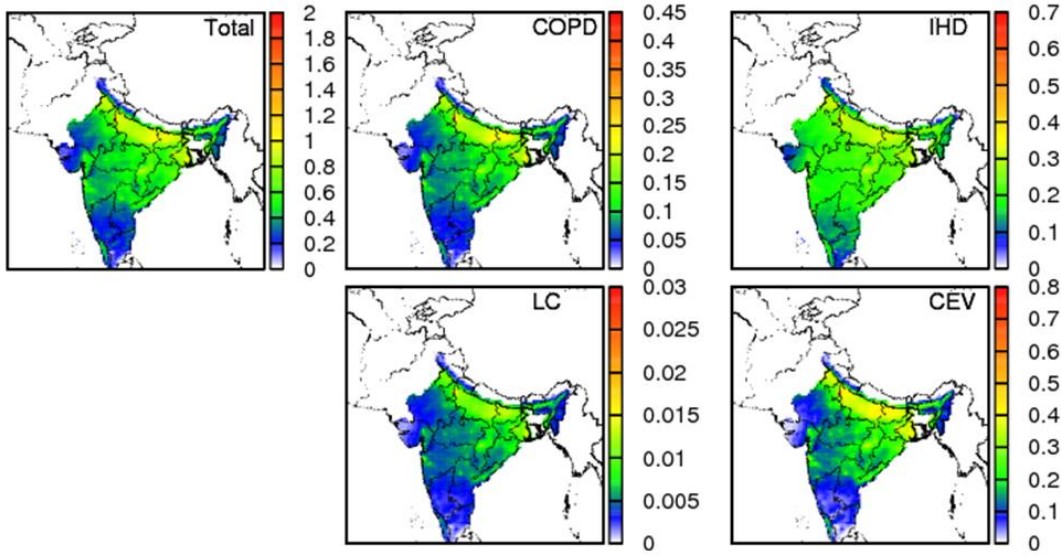

Figure 2. Year of life lost (YLL) based on population (years) due to COPD, LC, IHD, and CEV.




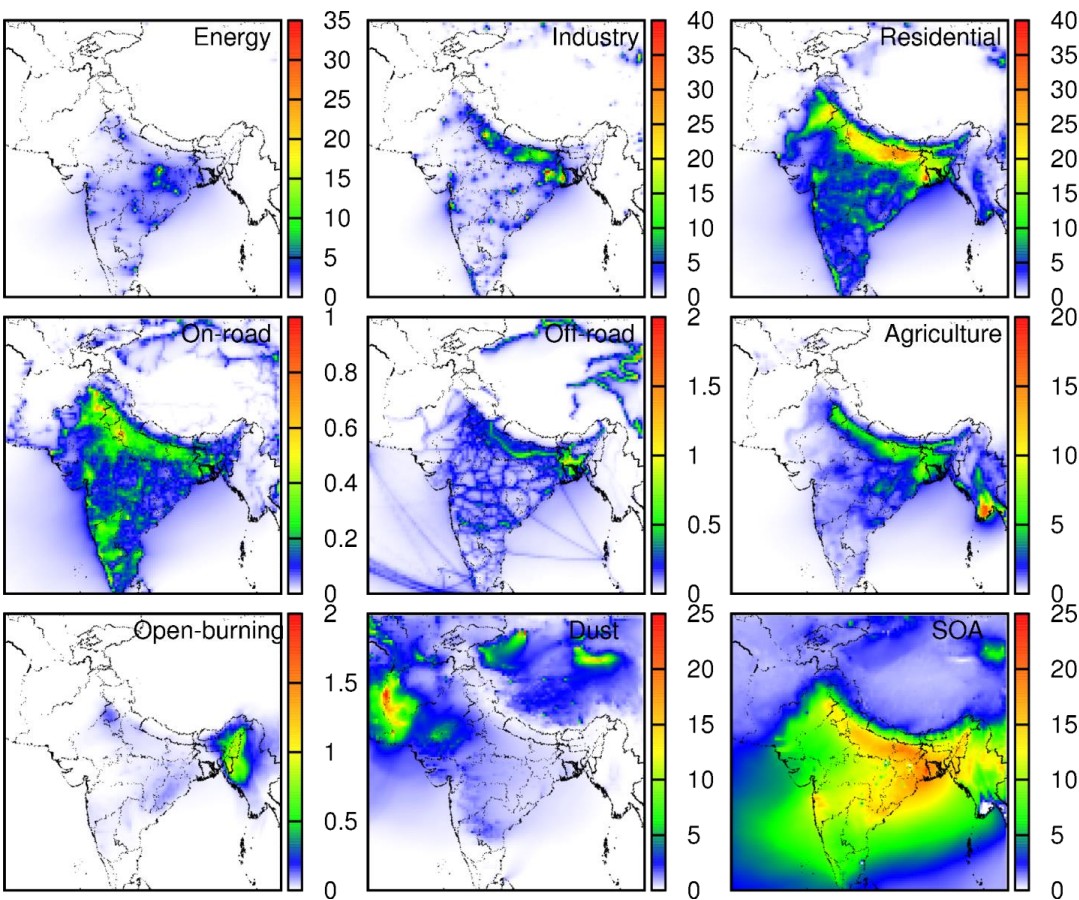

Figure 3. Source contributions to total PM$_{2.5}$ concentration (Units are in μg/m$^3$).



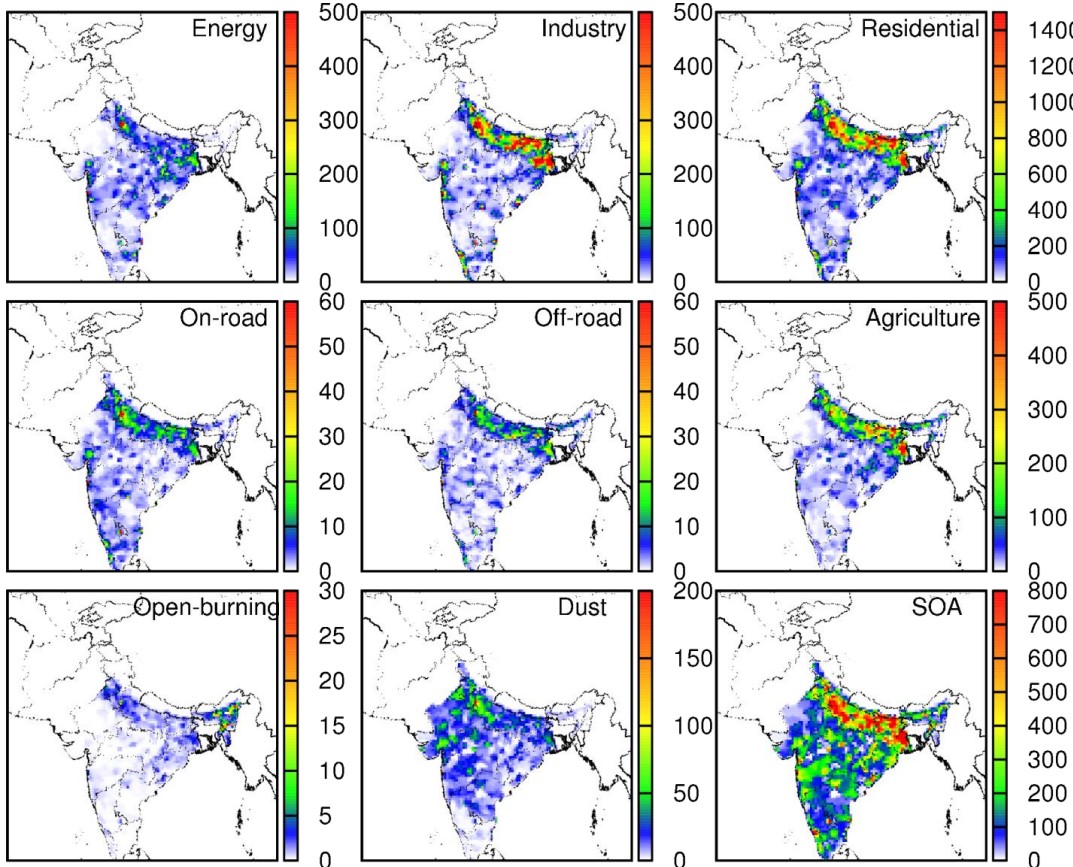

Figure 4. Source contributions to total premature mortality (deaths per grid $36 \times 36$ km) due to COPD, LC, IHD, and CEV.





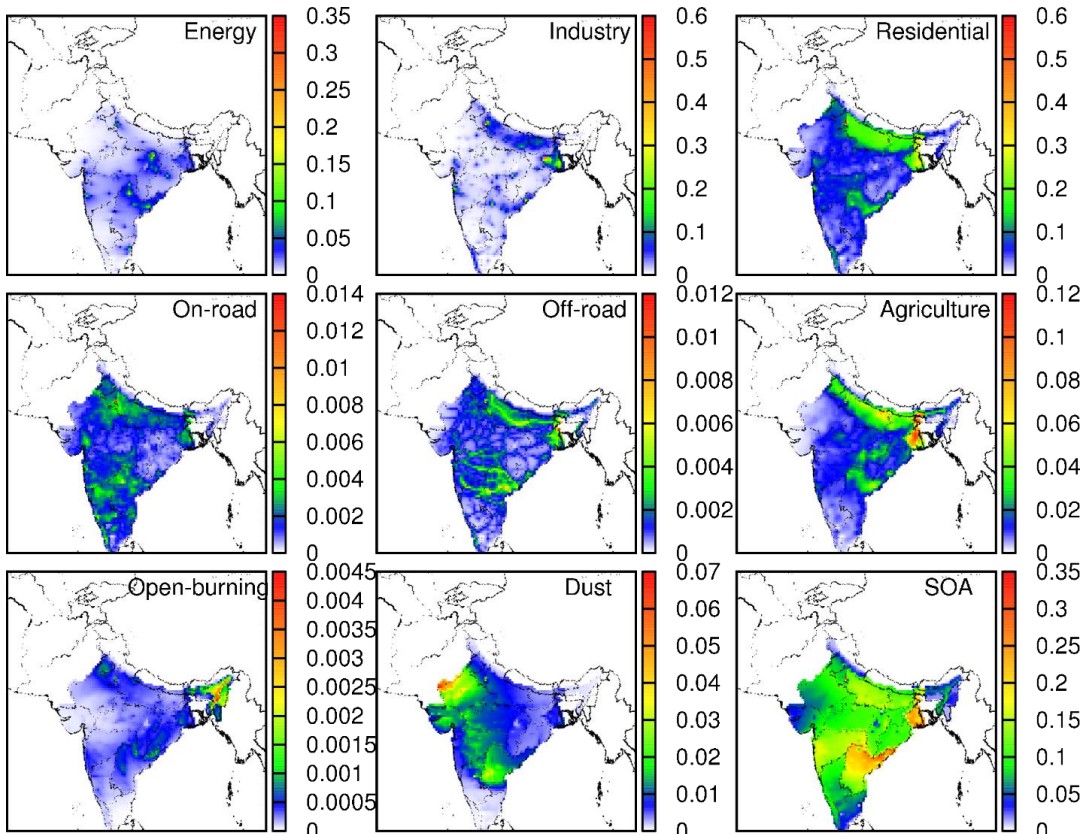

Figure 5. Contributions of different sources to years of life lost (YLL) based on population (years).



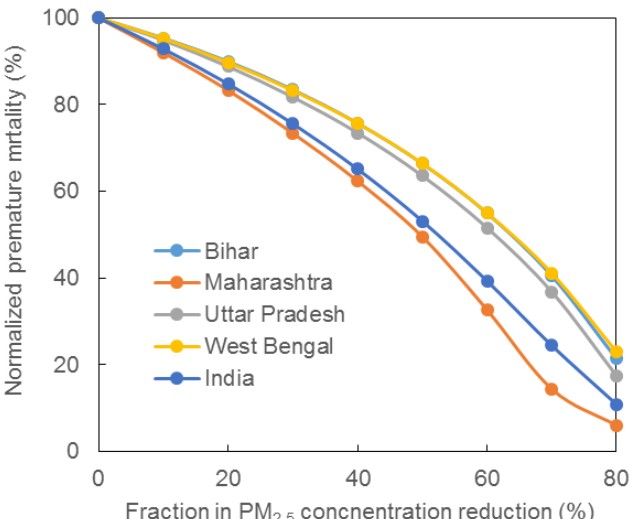

Figure 6. Premature mortality (normalized to 2015 deaths) as a function of the fractional reduction in PM$_{2.5}$ concentrations (relative to 2015 concentrations) for the whole of India and top PM$_{2.5}$ polluted states, Bihar, Maharashtra, Uttar Pradesh (including Delhi), West Bengal.




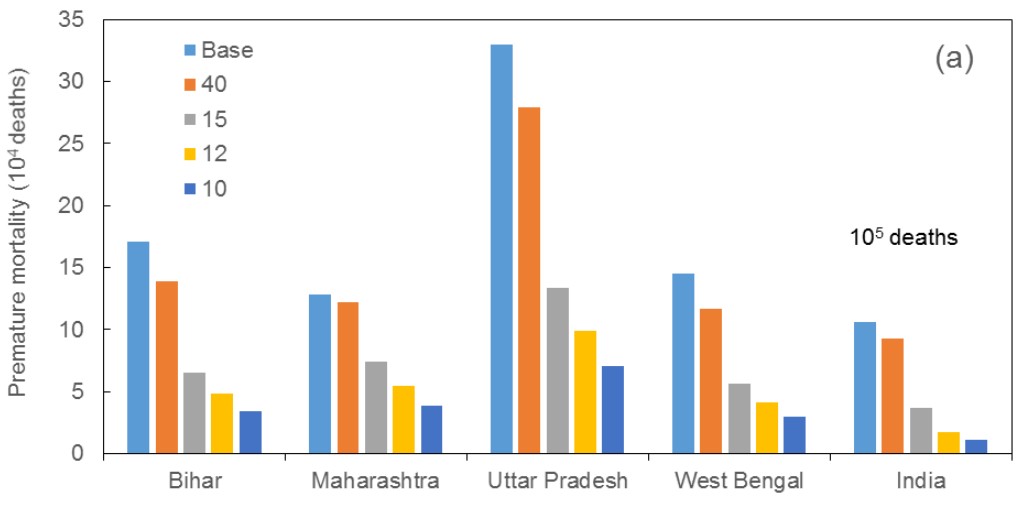

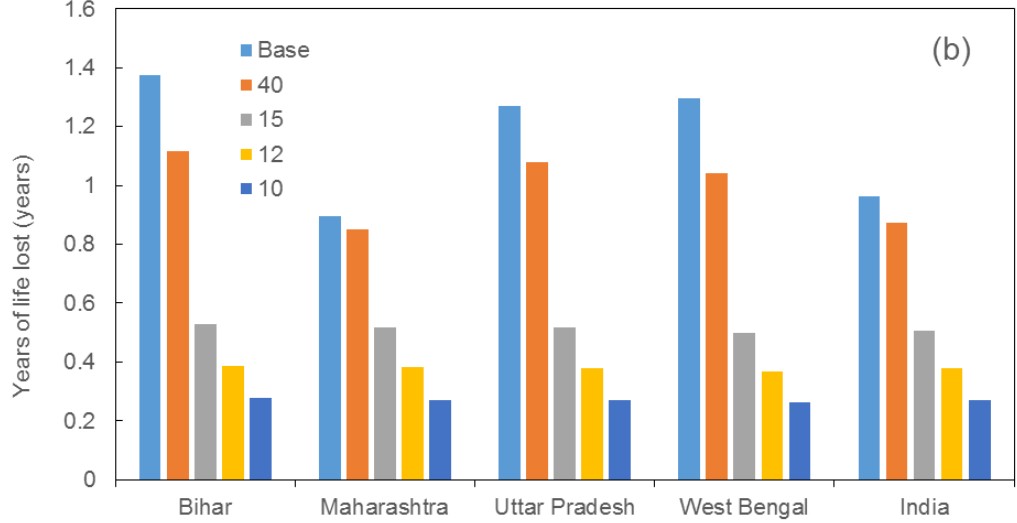

Figure 7. Number of premature deaths (a) and YLL (b) in the whole of India and top PM$_{2.5}$
polluted states, Bihar, Maharashtra, Uttar Pradesh (including Delhi) and West Bengal
corresponding to the cases when PM$_{2.5}$ reduced to 40μg/m$^3$, 15 μg/m$^3$, 12μg/m$^3$ and 10/μg m$^3$
(WHO guideline level). "Base" refers to PM$_{2.5}$ in 2015.