# Peer review of "Source contributions and potential reductions to health effects of particulate matter in India"

_Atmospheric Chemistry and Physics, 2018_

## Referee Comment (RC1) · Anonymous Referee #1 · 11 Jun 2018

Review for Atmospheric Chemistry and Physics Discussions

Title: Source contributions and potential reductions to health effects of particulate matter in India

Authors: Hao Guo, Sri Harsha Kota, Kaiyu Chen, Shovan Kumar Sahu, Jianlin Hu, Qi Ying, Yuan Wang, Hongliang Zhang

General comments

This manuscript estimates the contributions of different sources to ambient PM2.5 concentrations in India and the associated disease burden. The study calculates potential reductions in the health impacts if PM2.5 concentrations were reduced to different standards. The topic of Indian air quality is important as exposure to air pollution

causes a substantial disease burden in India and it is relevant to the scope of ACP. The author's use a regional chemical transport model at high-resolution to estimate the health impacts from ambient PM2.5 exposure with a methodology that is consistent with the literature, although they use outdated health functions and old baseline mortality data. The tagging methodology, using tracers to estimate the source contributions, is a strength of this study. The results are sufficient to support the conclusions that residential emissions dominate the source contributions, that the disease burden is primarily across northern India, and that large emission reductions are required to reduce the substantial disease burden from ambient PM2.5 exposure in India.

The major issue is the novelty of the manuscript. The authors state on line 76 and 77 that "no studies have attributed the health effects to different sources of PM2.5 in India till date". This is not true. The impacts of different sources to ambient PM2.5 concentrations and the associated disease burden in India were studied in detail in Lelieveld et al., (2015), Silva et al., (2016), Lelieveld (2017), Conibear et al., (2018a), GBD MAPS Working Group (2018), and Venkataraman et al., (2018). Only one out of six of these studies (Lelieveld et al., 2015) was discussed in this manuscript, and the results of this manuscript have largely been found in the other previous studies. Many studies have focused on reducing PM2.5 concentrations in India, for example, Giannadaki et al., (2016) studied the health impacts from applying different air quality standards to PM2.5 and Conibear et al., (2018b) explored the non-linear response of health impacts to PM2.5. The GBD MAPS Working Group (2018) and Venkataraman et al., (2018) directly addressed the research question of this manuscript studying source contributions and potential reductions of PM2.5 pollution in India in the present day and the future in comprehensive papers, one of which was recently published in ACP.

In summary, this manuscript focuses on an important topic using standard methods, though it neglects many previous studies that have already addressed this research question, and the current version of the manuscript is not novel. To develop the novelty of this manuscript, the author's could focus on the insights brought by the tagging

methodology relative to a zero-out approach and on the chemical speciation of PM2.5 health impacts seeing that SOA has a large impact in this work.

Specific comments

1. The author's should discuss the important work done on these research questions by Venkataraman et al., (2018), GBD MAPS Working Group (2018), Conibear et al., (2018a, 2018b), Lelieveld (2017), Silva et al., (2016), Giannadaki et al., (2016), GBD2016 (2017), Cohen et al., (2017), Chafe et al., (2014), and Butt et al., (2016).

2. Lines 46-49: Estimates that are more recent exist. In the GBD2016 (2017), India accounted for 1.034 million of 4.093 million global premature mortalities from ambient PM2.5 exposure, and ambient PM2.5 exposure was the second largest risk for health in India in 2016.

3. Line 54: "Few studies estimate the health effects using regional and global models, and satellite data". This is not true. More than 15 studies estimate the health effects using models and observations in India, where some are summarised in Figure 4a of Conibear et al., (2018).

4. Lines 61-65: The estimate of the disease burden from ambient PM2.5 exposure for the United States using a different health function is unrelated to this manuscript focusing on India.

5. The baseline mortality rates are for 2000. Large differences have occurred in these values relative to the year of study (2015).

6. The integrated exposure-response (IER) function used to calculate the health impacts uses coefficients from the GBD2010 (2012) study documented in Burnett et al., (2014). The IER has been updated multiple times (in 2013, 2015, and 2016). Estimates of the disease burden are very sensitive to the exposure-response function used and recent updates of the IER provide estimates that are more accurate.

7. Section 3.3: It is not clear how the reductions in PM2.5 and disease burden were

calculated.

8. The quality of the plots could be improved, e.g. increasing the resolution, not using a rainbow colour bar, adding units, and fixing typos (Figure 6).

9. The model evaluation should at least be summarised in this manuscript.

10. Line 191-192: Why does the approach to calculating YLL in Ghude et al., (2016) introduce uncertainties?

Technical corrections

1. The wording is sometimes unclear. Examples are Lines 58-61, 157, 189-192, 196-199, though this is not an exhaustive list.

2. Equations 3 and 4 could be consistent e.g., both include mortality.

3. Line 275: Typo "Utter Pradesh".

References

Burnett, R. T. et al. An integrated risk function for estimating the global burden of disease attributable to ambient fine particulate matter exposure. Environ. Health Perspect. 122, 397–403 (2014).

Butt, E. W. et al. The impact of emissions from residential combustion on atmospheric aerosol, human health and climate. Atmos. Chem. Phys. 16, 873–905 (2016).

Chafe, Z. A. et al. Household Cooking with Solid Fuels Contributes to Ambient PM2.5 Air Pollution and the Burden of Disease. Environ. Health Perspect. 122, 1314–1320 (2014).

Cohen, A. J. et al. Estimates and 25-year trends of the global burden of disease attributable to ambient air pollution: An analysis of data from the Global Burden of Diseases Study 2015. Lancet 389, 1907–1918 (2017).

Conibear, L., Butt, E. W., Knote, C., Arnold, S. R. & Spracklen, D. V. Residential energy

use emissions dominate health impacts from exposure to ambient particulate matter in India. Nat. Commun. 9, 9 (2018a).

Conibear, L., Butt, E. W., Knote, C., Arnold, S. R. & Spracklen, D. V. Stringent emission control policies can provide large improvements in air quality and public health in India. GeoHealth 2, (2018b).

GBD 2010 Risk Factors Collaborators. A comparative risk assessment of burden of disease and injury attributable to 67 risk factors and risk factor clusters in 21 regions, 1990-2010: a systematic analysis for the Global Burden of Disease Study 2010. Lancet 380, 2224–60 (2012).

GBD 2016 Risk Factors Collaborators. Global, regional, and national comparative risk assessment of 84 behavioural, environmental and occupational, and metabolic risks or clusters of risks, 1990–2016: a systematic analysis for the Global Burden of Disease Study 2016. Lancet 390, 1345–1422 (2017).

GBD MAPS Working Group. Burden of Disease Attributable to Major Air Pollution Sources in India. Special Report 21. Boston, MA:Health Effects Institute. (2018).

Ghude, S. D. et al. Premature mortality in India due to PM2.5 and ozone exposure. Geophys. Res. Lett. 43, 1–9 (2016).

Giannadaki, D. et al. Implementing the US air quality standard for PM2.5 worldwide can prevent millions of premature deaths per year. Environ. Heal. 15, 1–11 (2016).

Lelieveld, J., Evans, J. S., Fnais, M., Giannadaki, D. & Pozzer, A. The contribution of outdoor air pollution sources to premature mortality on a global scale. Nature 525, 367–371 (2015).

Lelieveld, J. Clean air in the Anthropocene. Faraday Discuss. 200, 693–703 (2017).

Silva, R. A., Adelman, Z., Fry, M. M. & West, J. J. The Impact of Individual Anthropogenic Emissions Sectors on the Global Burden of Human Mortality due to Ambient

Air Pollution. Environ. Health Perspect. 124, 1776–1784 (2016).

Venkataraman, C. et al. Source influence on emission pathways and ambient PM2.5 pollution over India (2015–2050). Atmos. Chem. Phys. 18, 8017–8039 (2018).

---

## Referee Comment (RC2) · Anonymous Referee #2 · 19 Jul 2018

This article studied the health effects of exposure to fine particulate matter in India using the source oriented CMAQ model. It quantified the premature mortality due to exposure to fine particulate matter in India based on CMAQ simulation of air quality for India in 2015. It also compared the mortality estimate with other existing studies. A new aspect of the study is that the source oriented CMAQ model allows it to quantify contributions to premature mortality from different source sectors. The residential sector was found to be the largest contributor. This can provide compelling argument for prioritizing emission control from that emission sector. In addition, it also estimated the health benefits if PM2.5 concentrations in India are reduced to levels corresponding to different air quality standards. Certainly, this highlighted the enormous health benefit from reduced PM2.5 concentrations. I found the findings of the article to be significant

and relevant for publication. I have the following comments for the authors to address.

1.) Lines 105-106, the article can provide more information about the model performance, particularly with regard to PM2.5 predictions in India. It will also strength the paper if it can provide any comments on source apportionment results (e.g., comparison to other published study or observations if possible)

2.) Since SOA is found to be significant contributor to PM2.5 and mortality, any comments on the sources of the SOA (e.g., biogenic or anthropogenic)?

3.) Table S3 should be moved to the main body of the paper. In addition, this table can provide more information about the difference in these studies (e..g, models used, emissions, resolution, mortality estimate method, etc.)

4.) Lines 103-104, is "open burning" referred later in the article corresponding to wildfires?

5.) Lines 113-120, what are the distribution assumptions used for Monte Carlo simulations?

6.) Line 137, a "." is missing after Table S2. "ai is the remaining years..." should be moved to line 137.

7.) A map showing the locations that are referred in the paper could be provided in supplemental material. This will help readers who are not familiar with geography of India.

8.) Table 1 could be revised. The states can be grouped to east India, north India, south, northeast, west, and central as discussed in lines 144-154.

9.) Lines 194-200, the description about source contributions is not clear and needs to be revised. It seems that the maximum contribution among grid cells is used to describe the significance of source contributions. Would average values or population weighted average values in India be more appropriate? Similarly, in the conclusion

and abstract part, this (e.g., 40 ug/m3 from residential sector) needs to be clear about whether it is maximum or average.

10.) Line 202, missing "are" after "power plants".

11.) Line 257, "for" changes to "of"

12.) Line 260, add "respectively" after "0.39 year".

13.) Line 273, similar to comment 9, the source contribution of ∼ 40 ug/m3 is just the maximum contribution among different grid cells, correct?
* * *

---

## Author Comment (AC1) · 10 Sep 2018

Dear Reviewer,

Thank you for the comments to help improve the quality of the paper. We have revised the manuscript to address your comments. A detailed response to each comment is provided below.

Anonymous Referee #2

This article studied the health effects of exposure to fine particulate matter in India using the source oriented CMAQ model. It quantified the premature mortality due to exposure to fine particulate matter in India based on CMAQ simulation of air quality for India in 2015. It also compared the mortality estimate with other existing studies. A

new aspect of the study is that the source oriented CMAQ model allows it to quantify contributions to premature mortality from different source sectors. The residential sector was found to be the largest contributor. This can provide compelling argument for prioritizing emission control from that emission sector. In addition, it also estimated the health benefits if PM2.5 concentrations in India are reduced to levels corresponding to different air quality standards. Certainly, this highlighted the enormous health benefit from reduced PM2.5 concentrations. I found the findings of the article to be significant and relevant for publication. I have the following comments for the authors to address.

Response: The authors thank the reviewer for the positive comments and addressed below comments carefully.

1.) Lines 105-106, the article can provide more information about the model performance, particularly with regard to PM2.5 predictions in India. It will also strength the paper if it can provide any comments on source apportionment results (e.g., comparison to other published study or observations if possible) Responses: Summarized validation results and discussions of comparison with other source apportionment study were added. Changes in manuscript: Lines 110 to 117 were added in the revised manuscript.

2.) Since SOA is found to be significant contributor to PM2.5 and mortality, any comments on the sources of the SOA (e.g., biogenic or anthropogenic)? Responses: Figure S4, which showed components concentration of SOA, was added in supplemental materials and discussions were added. Changes in manuscript: Figure S4 was added and lines 215 to 216 were added.

3.) Table S3 should be moved to the main body of the paper. In addition, this table can provide more information about the difference in these studies (e..g, models used, emissions, resolution, mortality estimate method, etc.) Responses: Thanks for the suggestion. We added more information of other studies and moved TableS3 to Table 2. Changes in manuscript: Table S3 was moved to Table 2 in main draft. Discussions

were added in lines 237 to 240.

4.) Lines 103-104, is "open burning" referred later in the article corresponding to wildfires? Responses: Yes, it is wildfire. Sorry for the confusion. We modified line 108 to make it clear. Changes in manuscript: Statement "which is assigned as open-burning sector" was added to line 108.

5.) Lines 113-120, what are the distribution assumptions used for Monte Carlo simulations? Responses: Bayesian MCMC nonlinear curve-fitting was used by Global Burden of Disease (http://ghdx.healthdata.org/), where we could get the MC simulation results. Changes in manuscript: No changes.

6.) Line 137, a "." is missing after Table S2. "ai is the remaining years. . ." should be moved to line 137. Responses: Thanks for the correction. Modified as above. Changes in manuscript: Lines 148 to150 were modified.

7.) A map showing the locations that are referred in the paper could be provided in supplemental material. This will help readers who are not familiar with geography of India. Responses: Thanks for the suggestion. Changes in manuscript: Figure S2 was added to supplemental materials.

8.) Table 1 could be revised. The states can be grouped to east India, north India, south, northeast, west, and central as discussed in lines 144-154. Responses: As we also discussed some heavy-polluted states of India, Table 1 was kept. Changes in manuscript: No changes.

9.) Lines 194-200, the description about source contributions is not clear and needs to be revised. It seems that the maximum contribution among grid cells is used to describe the significance of source contributions. Would average values or population weighted average values in India be more appropriate? Similarly, in the conclusion and abstract part, this (e.g., 40 ug/m3 from residential sector) needs to be clear about whether it is maximum or average. Responses: Thanks for the suggestion. In order to

look at the spatial distribution and some hotspot on the map, we used maximum contribution among grid cells here. We modified the description to make it clear. Changes in manuscript: Lines 207 to 208 were modified.

10.) Line 202, missing "are" after "power plants". Responses: Thanks for pointing out. Changes in manuscript: Modified.

11.) Line 257, "for" changes to "of" Responses: Thanks for the correction. Changes in manuscript: Modified.

12.) Line 260, add "respectively" after "0.39 year". Responses: Thanks for the correction suggestion. Changes in manuscript: Added.

13.) Line 273, similar to comment 9, the source contribution of âĹij 40 ug/m3 is just the maximum contribution among different grid cells, correct? Responses: Yes, it is maximum contribution. Changes in manuscript: Line 288 was modified to "with source contribution of $\sim$ 40 $\mu$g/m3 maximum to total PM2.5".

Please also note the supplement to this comment:
https://www.atmos-chem-phys-discuss.net/acp-2018-483/acp-2018-483-AC1-supplement.pdf

—————————————————————

---

## Author Comment (AC2) · 11 Sep 2018

Dear Reviewer,

Thank you for the comments to help improve the quality of the paper. We have revised the manuscript to address your comments. A detailed response to each comment is provided below.

General comments: This manuscript estimates the contributions of different sources to ambient PM2.5 concentrations in India and the associated disease burden. The study calculates potential reductions in the health impacts if PM2.5 concentrations were reduced to different standards. The topic of Indian air quality is important as exposure to air pollution causes a substantial disease burden in India and it is relevant to the

[Figure]

scope of ACP. The author's use a regional chemical transport model at high-resolution to estimate the health impacts from ambient PM2.5 exposure with a methodology that is consistent with the literature, although they use outdated health functions and old baseline mortality data. The tagging methodology, using tracers to estimate the source contributions, is a strength of this study. The results are sufficient to support the conclusions that residential emissions dominate the source contributions, that the disease burden is primarily across northern India, and that large emission reductions are required to reduce the substantial disease burden from ambient PM2.5 exposure in India. The major issue is the novelty of the manuscript. The authors state on line 76 and 77 that "no studies have attributed the health effects to different sources of PM2.5 in India till date". This is not true. The impacts of different sources to ambient PM2.5 concentrations and the associated disease burden in India were studied in detail in Lelieveld et al., (2015), Silva et al., (2016), Lelieveld (2017), Conibear et al., (2018a), GBD MAPS Working Group (2018), and Venkataraman et al., (2018). Only one out of six of these studies (Lelieveld et al., 2015) was discussed in this manuscript, and the results of this manuscript have largely been found in the other previous studies. Many studies have focused on reducing PM2.5 concentrations in India, for example, Giannadaki et al., (2016) studied the health impacts from applying different air quality standards to PM2.5 and Conibear et al., (2018b) explored the non-linear response of health impacts to PM2.5. The GBD MAPS Working Group (2018) and Venkataraman et al., (2018) directly addressed the research question of this manuscript studying source contributions and potential reductions of PM2.5 pollution in India in the present day and the future in comprehensive papers, one of which was recently published in ACP. In summary, this manuscript focuses on an important topic using standard methods, though it neglects many previous studies that have already addressed this research question, and the current version of the manuscript is not novel. To develop the novelty of this manuscript, the author's could focus on the insights brought by the tagging methodology relative to a zero-out approach and on the chemical speciation of PM2.5 health impacts seeing that SOA has a large impact in this work.

[Figure]

Responses: We thank the reviewer for all the suggestions, which are helpful to improve the manuscript. We modified the introduction to add more discussion of previous researches, highlighted the novelty of this study and addressed below specific comments. a) We are sorry for missing new references while we prepared the manuscript. Now all the six studies are now discussed in the Introduction section at lines 56 to 62. Please be noted that Lelieveld (2017) shows the same values as Lelieveld (2015). "The impacts of different sources on ambient PM2.5 concentrations and the associated disease burden in global scale were also studied in Silva et al. (2016) and Lelieveld (2017). Giannadaki et al. (2016) and Conibear et al. (2018) studied the health impacts from applying different air quality standards and explored the non-linear response of health impacts to PM2.5 in India. The GBD MAPS Working Group (2018) and Venkataraman et al. (2018) focused on source contributions and potential reductions of PM2.5 in India in the present day and the future using the brute force method by removing certain sources". b) Although these studies have investigated different aspects of health effects from different sources or benefits from potential reductions, they have not addressed the questions answered in this study, which highlights the novelty and merit of this study. The comparison of the methods and results of this study with previous studies is included in Table 2. a. First, this study uses the tagged tracer method, which is not affected by the non-linearity of atmospheric processes. Other studies all used brute force (i.e., zero-out) method if they did source apportionment, which changed the atmospheric processes and caused potential uncertainties. For example, reducing emission of PM would change the transport, deposition, surface related reactions, and reducing emissions of NOx and VOCs would change the formation of photochemical pollutants such as ozone and SOA. b. The health analysis of this study is based on modified CMAQ with improved performance on PM based on companion papers (Kota et al., 2014, Kota et al., 2015; Ying et al., 2015; Zhang and Ying, 2010). This study also has better spatial resolution compared to global studies and similar resolution compared to India centered studies. c. The study is more comprehensive in understanding the health effects and benefits of concentration reductions

of PM2.5. We estimated the deaths caused by different diseases (only Lelieveld et al., 2015 and Silva et al., 2016) and different sources (Lelieveld et al., 2015, Conibear et al., 2018, GBD MAPS Working Group 2018 and Venkataraman et al., 2018 did), we estimated years of life lost in addition to mortality (only Ghude et al., 2016 did), and we estimated the potential benefits of PM2.5 reductions (only Giannadaki et al., 2016 and Conibear et al., 2018 did). It should be noted that all these are based on improved CMAQ performance and tagged tracer method. Thus, we believe our manuscript has its novelty and merit, and contribute to the understanding of air pollution in India. We did not add comparison of tagged tracer method and the brute force method, although it is a good suggestion, because it does not fall in the focus of the study. I believe it is significant as it has been shown and discussed in many studies worldwide. The health impacts of chemical speciation of PM2.5 are another good idea, however it is not doable because we are missing the concentration-response functions for the components. You can get results if use same functions as total PM2.5, but it is not meaningful. This should also be the reason that why no studies did this, although they all have the components information from their models. We modified lines 77 to 80 to be clearer about the merits of this study as below: "Although previous studies have addressed different aspects of health impact of PM2.5 in India, a comprehensive understanding on source contributions and potential reductions to both premature mortality and YLL using a tagged tracer method with updates to better predict PM2.5 in India is missing".

Specific comments

1. The author's should discuss the important work done on these research questions by Venkataraman et al., (2018), GBD MAPS Working Group (2018), Conibear et al., (2018a, 2018b), Lelieveld (2017), Silva et al., (2016), Giannadaki et al., (2016), GBD2016 (2017), Cohen et al., (2017), Chafe et al., (2014), and Butt et al., (2016). Responses: We added discussions of all these papers. Changes in manuscript: Lines 56 to 62 were added in the revised manuscript.

2. Lines 46-49: Estimates that are more recent exist. In the GBD2016 (2017), India

accounted for 1.034 million of 4.093 million global premature mortalities from ambient PM2.5 exposure, and ambient PM2.5 exposure was the second largest risk for health in India in 2016 Responses: Thanks for the most recent data. We added the GBD estimates after the Lelieveld et al. discussion. The statements in lines 43 to 45 were modified. Changes in manuscript: Lines 43 to 45 now read "In the Global Burden of Disease Study 2016 (GBD, 2017), India accounted for 1.034 million of 4.093 million global premature mortalities from ambient PM2.5 exposure, and ambient PM2.5 exposure was the second largest risk for health in India".

3. Line 54: "Few studies estimate the health effects using regional and global models, and satellite data". This is not true. More than 15 studies estimate the health effects using models and observations in India, where some are summarized in Figure 4a of Conibear et al., (2018). Responses: Modified. Changes in manuscript: Line 52 now read "Several studies have estimated the health effects using regional and global models, and satellite data".

4. Lines 61-65: The estimate of the disease burden from ambient PM2.5 exposure for the United States using a different health function is unrelated to this manuscript focusing on India. Responses: Removed as suggested. Changes in manuscript: This sentence was now removed.

5. The baseline mortality rates are for 2000. Large differences have occurred in these values relative to the year of study (2015). Responses: There was a tyro here. The baseline mortality rates are for 2010 as the most recent data we can find. Changes in manuscript: Corrected tyro to 2010.

6. The integrated exposure-response (IER) function used to calculate the health impacts uses coefficients from the GBD2010 (2012) study documented in Burnett et al., (2014). The IER has been updated multiple times (in 2013, 2015, and 2016). Estimates of the disease burden are very sensitive to the exposure-response function used and recent updates of the IER provide estimates that are more accurate. Responses:

Thanks for the suggestion. There were several IER functions used in previous studies. Recent India health studies like Giannadaki et al., (2016) and Conibear et al., (2018b) were all based on the IER function in Burnett et al., (2014), so we used the same to make our studies comparable with other studies. Changes in manuscript: No changes.

7. Section 3.3: It is not clear how the reductions in PM2.5 and disease burden were calculated. Responses: The reduction of PM2.5 was calculated by original PM2.5 concentration time reduction fraction. The mortality was then calculated using PM2.5 concentration after reduction. Changes in manuscript: Description was added to lines 248 to 249.

8. The quality of the plots could be improved, e.g. increasing the resolution, not using a rainbow colour bar, adding units, and fixing typos (Figure 6). Responses: The figures were renewed now. Fixed typos. The rainbow color can present the spatial distribution better, so we did not modify here. Changes in manuscript: Figures renewed. Fixed typo in Figure 6.

9. The model evaluation should at least be summarized in this manuscript. Responses: Summarized validation results are added. Changes in manuscript: Lines 110 to 114 were added in the revised manuscript.

10. Line 191-192: Why does the approach to calculating YLL in Ghude et al., (2016) introduce uncertainties? Responses: They are using a linear relationship assumption that an increase of 1 $\mu$g/m3 in PM2.5 exposure decreases mean life expectancy by about 0.061±0.02 years, but the relationship between YLL and PM2.5 should be nonlinear. Changes in manuscript: Lines 201 to 202 were modified as above.

Technical corrections

1. The wording is sometimes unclear. Examples are Lines 58-61, 157, 189-192, 196-199, though this is not an exhaustive list. Responses: Sorry for the confusion. The above lines were modified and we went through the whole draft again to avoid confusion. Changes in manuscript: Lines 63 to 65, 167 to 168, 201 to 202, and 207 to 208 were modified.

2. Equations 3 and 4 could be consistent e.g., both include mortality. Responses: Eq.3 and 4 are now consistent. Changes in manuscript: Eq.3 and 4 were modified.

3. Line 275: Typo "Utter Pradesh". Responses: Sorry for the tyro. We corrected it. Changes in manuscript: Modified.

Please also note the supplement to this comment:
https://www.atmos-chem-phys-discuss.net/acp-2018-483/acp-2018-483-AC2-supplement.pdf

─────────────────────────────

---

## Author Response (AR2)

Dear Editor,

Thank you for the minor comments to improve the quality of the paper. We have revised the manuscript based on your comments. A detailed response to each comment is provided in this file with comments from comments in black, author's response in red, and author's changes in manuscript in blue.

1. Table 2 is a useful comparison of previous studies and helps highlight the novelty of this work. Please double check the values entered in this table. For example, the Excess Mortality value for Conibear et al. (2018) is not correct. It should be 0.99 million (see Page 3 of Conibear et al.). Please check the numbers from the other studies as well.
Responses: Sorry for the mistake. We have checked through all the mentioned studies and ensured all the numbers are correct now. We also updated Table 2 with more information from other studies and described them in related parts in the manuscript.
Changes in manuscript: Corrected errors and added more information in Table 2.

2. Line 185-Line 191. This paragraph should be updated to include the other studies now included in Table 2.
Responses: Thanks for comments and we added all the discussion towards the studies mentioned in Table at lines 191 to 200.
Changes in manuscript: Lines 191 to 200 were updated with more discussions in Table 2.

3. Line 160 and Table 2. It would be useful to compare population weighted PM2.5 concentrations from previous studies when this is available.
Responses: We only found annual population weighted $PM_{2.5}$ concentration discussed in Conibear et al. (2018) and GBD MAPS Working Group (2018) and added them to Table 2. Due to the differences in simulation period and resolutions, the values are much different. The following discussions were added in Lines 161 to 163.
Changes in manuscript: Population weighted $PM_{2.5}$ concentrations in Conibear et al. (2018) and GBD MAPS Working Group (2018) were added to Table 2 and following discussions were added to lines 163 to 165.